# Evaluation of Nitrogen Oxide Reduction Performance in Permeable Concrete Surfaces Treated with a TiO_2_ Photocatalyst

**DOI:** 10.3390/ma16165512

**Published:** 2023-08-08

**Authors:** Hyeok-jung Kim, Kinam Hong

**Affiliations:** 1Industry Academic Cooperation Foundation, Hankyong National University, 327 Jangang-ro, Anseong 17579, Republic of Korea; ceasare@hknu.ac.kr; 2Department of Civil Engineering, Chungbuk National University, 1 Chungdae-ro, Seowon-Gu, Cheongju 28644, Republic of Korea

**Keywords:** ecofriendly permeable concrete, photocatalysis, fine dust precursor, air pollutant degradation, nitrate

## Abstract

Fine dust, recently classified as a carcinogen, has raised concerns about the health effects of air pollution. Vehicle emissions, particularly nitrogen oxide (NO_x_), contribute to ultrafine dust formation as a fine dust precursor. A photocatalyst, such as titanium dioxide (TiO_2_), is a material that causes a catalytic reaction when exposed to light, has exceptional characteristics such as decomposition of pollutants, and can be used permanently. This study aimed to investigate NO_x_ reduction performance by developing ecofriendly permeable concrete with photocatalytic treatment to reduce fine dust generated from road mobile pollution sources. Permeable concrete specimens containing an activated loess and zeolite admixture were prepared and subjected to mechanical and durability tests. All specimens, including the control (CTRL) and admixture, met quality standard SPS-F-KSPIC-001-2006 for road pavement. Slip resistance and permeability coefficient also satisfied the standards, while freeze–thaw evaluation criteria were met only by CTRL and A1Z1 specimens. NO_x_ reduction performance of the permeable concrete treated with TiO_2_ photocatalyst was assessed using ISO standard and tank chambers. NO_x_ reduction efficiency of up to 77.5% was confirmed in the permeable concrete specimen with TiO_2_ content of 7.5%. Nitrate concentration measurements indirectly confirmed photolysis of nitrogen oxide. Incorporating TiO_2_ in construction materials such as roads and sidewalks can improve the atmospheric environment for pedestrians near roads by reducing NO_x_ levels through photocatalysis.

## 1. Introduction

Air pollution due to rapid economic growth is one of the problems facing the world, mainly in the form of fine dust in the process of burning fossil fuels such as coal and oil in factories and automobiles, etc. [1]. Particulate matter (PM) refers to atmospheric substances that exist in the form of particles. It is generally classified as fine dust (PM_10_) when the diameter of the largest particle is below 10 μm or as ultrafine dust (PM_2.5_) when the diameter of the largest particle is below 2.5 μm. In particular, PM can easily infiltrate bronchial tubes and lungs; therefore, it causes various respiratory diseases as well as cardiovascular, skin, and eye diseases. Samet et al. (2000) reported that the total mortality rate increases by 0.51% if the concentration of atmospheric PM_10_ increases [2]. Other studies have reported that the total mortality rate could increase up to 4–18% upon long-term exposure to PM [3,4]. Therefore, the International Agency for Research on Cancer (IARC), an organization affiliated with the World Health Organization (WHO), has classified PM as a Group 1 carcinogen [5], and measures for reducing atmospheric PM are urgently needed.

Nitrogen oxide (NO_x_) accounts for approximately 60% of PM originating from the roadside. NO_x_ is a pollution source that secondarily generates PM_2.5_ by reacting with vapor (H_2_O), ozone (O_3_), and organic compounds under certain conditions in the atmosphere. NO_x_ is mainly generated by the combustion of fossil fuels such as coal or petroleum. In particular, it is estimated that a significant amount of NO_x_ is emitted from road mobility pollution sources in large cities with significant vehicle transit, and it has been reported that approximately 45% of NO_x_ from Seoul has been emitted through road mobility pollution sources [6]. Therefore, for reducing NO_x_ in urban areas, controlling NO_x_ emitted through road mobility sources or using road facilities is effective. Among various road facilities, those composed of concrete, such as boundary stones, bagged and sacked concrete, and crash walls, have large surface areas; therefore, the use of infrastructure composed of functional concrete is an efficient method of reducing roadside NO_x_ [7].

TiO_2_ photocatalysts, which are most commonly used as functional materials for reducing NO_x_ as a precursor to PM, can be used semi-permanently because it does not change under light, and it is mainly included in concrete, asphalt mixture, paint for mortar, coating materials, and packing materials owing to its excellent capability to decompose pollutants [8,9,10]. TiO_2_ photocatalysts have a 3.0–3.2 eV band gap, and electrons (e^−^) and electron holes (h^+^) are formed when TiO_2_ is irradiated using light with a wavelength of 380 nm (ultraviolet rays) or below. The generated e^−^ and h^+^ are strong reducing agents and oxidizing agents, respectively; thus, hydroxyl radical (OH) and superoxide anion (O_2_^−^) can be created through reaction with water and oxygen from the surrounding atmosphere, demonstrating excellent efficiency for decomposing atmospheric pollutants [11]. The NO_x_ decomposition reaction mediated by TiO_2_ is presented in Equation (1), and Figure 1 shows a schematic representation of the NO_x_ decomposition process [12].
(1)$$\begin{array}{c} \mathrm{Activation}\;:{\mathrm{TiO}}_{2}+{\mathrm{hv}}^{*}\to {\;h}^{+}+e^{-}\\ \mathrm{Absorption}\;:{\;H}_{2}O+{\mathrm{Site}}^{**}\to H_{2}O_{\mathrm{ads}}\\ \mathrm{Hole}\;\mathrm{trapping}\;:{\;O}_{2}\left(g\right)+{\mathrm{Site}}^{**}\to O_{2\mathrm{ads}}\\ \;\mathrm{NO}\left(g\right)+{\mathrm{Site}}^{**}\to {\mathrm{NO}}_{\mathrm{ads}}\\ {\mathrm{NO}}_{2}\left(g\right)+{\mathrm{Site}}^{**}\to {\mathrm{NO}}_{2\mathrm{ads}}\\ \mathrm{Electrontrapping}\;:{\;O}_{2}+e^{-}\to O_{2}^{-}\\ \mathrm{Hydroxyl}\;\mathrm{attack}\;:{\;\mathrm{NO}}_{\mathrm{ads}}+2\mathrm{OH}\cdot \;\to +{\mathrm{Site}}^{**}\to {\mathrm{NO}}_{2\mathrm{ads}}+H_{2}O\\ \;{\mathrm{NO}}_{2\mathrm{ads}}+\mathrm{OH}\cdot \;\to {\mathrm{HNO}}_{3}^{-}+H^{+}\\ {\mathrm{hv}}^{*}\;:\mathrm{UV},{\mathrm{Site}}^{**}:{\;\mathrm{Surface}\;\mathrm{of}\;\mathrm{TiO}}_{2} \end{array}$$

Various studies have investigated the reduction in atmospheric NO_x_ using TiO_2_ photocatalysts that demonstrate excellent decomposition of organic matter. Guo et al. (2020) assessed the photocatalytic decomposition efficiency of concrete consisting of nano-TiO_2_ photocatalysts [13]. They utilized two methods for integrating nano-TiO_2_ into the concrete: (1) directly mixing nano-TiO_2_ particles with concrete and (2) spreading nano-TiO_2_ particles onto the concrete surface by spraying. They reported that the photocatalytic efficiency improved when concrete was not polished according to the increase in nano-TiO_2_ content increase as well as the increase in the concentration of pollutants. Gopala Krishna Sastry et al. (2021) experimentally studied the effect of nano-TiO_2_ on fly ash-based geopolymer concrete durability and strength [14]. In their research, the compressive strength, bending strength, and splitting tensile strength of geopolymer concrete in which fly ash was substituted with nano-TiO_2_ were assessed; the fly ash-based geopolymer concrete was soaked in magnesium sulfate and sodium chloride to assess durability. They reported that the strength and durability of the fly ash-based geopolymer concrete increased as the amount of substituted TiO_2_ increased. Kim et al. (2018) investigated the NO_x_ removal efficiency of the highway-retaining wall of South Korea’s Gyeongbu Expressway, where TiO_2_ was spread onto the concrete surface through a surface penetrant [15]. They reported that the NO_x_ concentration was reduced by approximately 12% following the addition of TiO_2_. Additionally, they reported that the NO_x_ removal efficiency increased as the amount of sunshine and traffic increased. Beeldens (2014) investigated the photocatalyst block pavement in Antwerp, Belgium [16], and Th. Maggos et al. (2007) performed an onsite experiment by applying TiO_2_ photocatalyst paint in a parking lot [17]. Furthermore, Gian Lsuca Guerrini (2012) used cement-based paint treated with TiO_2_ in the Umberto 1 tunnel in Rome to identify its NO_x_ reduction capacity [18]. On the other hand, since the activity of the TiO_2_ photocatalyst is only active in UV light, which is about 5% of sunlight [19,20], various methods have been studied to improve the activity, such as bonding with other semiconductors, noble metal loading, doping, and heterojunction configuration [21,22,23]. Islam Ibrahim et al. (2022) reported that the component combination of TiO_2_/g-C3N4@Ag NPs showed visible light activation and improved photocatalytic performance under not only UV but also visible light irradiation [24].

Thus, this study investigated the NO_x_ reduction performance of road pavement concrete coated with TiO_2_ photocatalysts to reduce fine dust generated from roadsides for pedestrians. Unlike asphalt or general concrete pavement, rainwater is drained to prevent aquaplaning, enabling smooth passage of vehicles and moving objects, and a permeable concrete containing ecofriendly materials was developed. In addition, a tank photoreactor was fabricated and the NO_x_ reduction efficiency of the specimen that utilized real-size construction material was determined. For this purpose, the mixing ratio of permeable concrete for sidewalk pavement was derived, and the compressive strength, bending strength, freeze–thawing resistance, skid resistance, and permeability performance were determined. Next, TiO_2_ photocatalyst was coated onto the fabricated permeable concrete mixed with the derived optimal mixing ratio to assess the NO_x_ reduction efficiency according to the TiO_2_ content. NO_x_ reduction assessment involved analysis using a newly developed tank chamber that can accommodate large-scale specimens and an ISO standard chamber that uses small specimens. Additionally, a water quality analyzer was used to identify the generated nitrate concentration after NO_x_ reduction assessment.

## 2. Materials and Methods

Most road pavements are made up of asphalt or concrete; hence, the surface layer and base layer are impervious. Consequently, rainwater wells up when there is heavy rainfall, resulting in hydroplaning. Hydroplaning reduces skid resistance, threatens safety, and disturbs smooth transit. Permeable concrete has increased permeability owing to the mixture of single-size particle aggregate, cement, and water. Paving roads with this permeable concrete can smoothly drain rainwater unlike general concrete/asphalt pavements and facilitate smooth transit of moving objects. In particular, using permeable concrete where TiO_2_ photocatalyst is mixed/spread as pavement material for sidewalks and bicycle roads reduces NO_x_ generated from road mobility pollution sources such as vehicles/motorcycles, and the effect of NO_x_ on pedestrians can be directly reduced. To this end, the durability of permeable concrete containing a mixture of active loess and zeolite and the NO_x_ reduction efficiency of TiO_2_ spray coating was assessed at the laboratory level.

### 2.1. Materials

#### 2.1.1. Permeable Concrete

Table 1 shows the chemical composition of cement for the mixture of permeable concrete for road pavement. The cement used in precast concrete pavement (PCP) is a Type 1 Portland cement fabricated in South Korea with a density of 3.15 g/m^3^ and Blaine of 3000 cm^2^/g.

The active loess is red clay that activates SiO_2_ and Al_2_O_3_ by rapidly cooling natural red clay after high-temperature heating. According to previous research, concrete containing active loess is known to have strong acid resistance. In particular, active loess as a material that forms a porous structure is an environmentally friendly material that has excellent far-infrared radiation emissivity and deodorant performance. Zeolite is mainly composed of SiO_2_ and Al_2_O_3_ similar to other pozzolan material and forms a porous structure similar to active loess. Zeolite is known to have outstanding absorption performance owing to its excellent cation exchange properties. Therefore, this study used porous active loess and zeolite as mixing materials for the permeable concrete to improve the coating performance of TiO_2_ photocatalyst. Figure 2 shows the active loess and zeolite used in this study, and Table 2 shows the chemical composition of active loess and zeolite provided by the suppliers. The coarse aggregate used in this study was a single-size particle thick aggregate with a diameter of 10 mm or above and 13 mm or below. The density and absorption rate of this aggregate were 2.34 g/m^3^ and 1.27%, respectively.

#### 2.1.2. Photocatalyst

The photocatalyst used as surface coating material of permeable concrete in this study was a product of AERODISP^®^W740X by Evonik Industries AG, and an anatase-type TiO_2_ with outstanding photolytic efficiency was used. The characteristics of AERODISP^®^W740X used in this study are summarized in Table 3.

### 2.2. Experimental Variable

Table 4 shows the mixing ratio of permeable concrete used in this study. The water–binder rate used in the mixture was 36.4%. Previous studies have reported that when the rate of substitution of porous pozzolan material for cement exceeds 20%, the strength of permeable concrete significantly decreases [25,26]. Therefore, the rate of substitution of porous pozzolan material for cement was set to 15%. As an experimental variable, several mixture ratios (2:1, 1:1, and 1:2) of active loess and zeolite were considered. Among the test specimen names in Table 4, CTRL refers to the permeable concrete in which porous material was not substituted, and the Arabic numbers behind A and Z refer to the mixture ratio of active loess and zeolite, respectively. For example, A2Z1 indicated a specimen where 10% of cement was substituted with active loess and 5% was substituted with zeolite.

The AERODISP^®^W740X product used as a surface coating material for permeable concrete has a high solid fraction with 40% TiO_2_ content. To determine the optimal TiO_2_ content for reducing NO_x_, photocatalysts with varying TiO_2_ content were used as shown in Table 5. Distilled water was used to dilute AERODISP^®^W740X. T0 refers to the permeable concrete where the photocatalyst was not spread.

### 2.3. Characterization

#### 2.3.1. Mechanical and Durability Tests

In the SPS-F-KSPIC-001-2006(2018) standard (Korea standard), the evaluation criteria and quality required performances of permeable concrete for road pavement are presented as shown in Table 6 [27]. In this standard, the performance of permeable concrete is differentiated according to the location of the road pavement (sidewalk, bicycle road, and parking lot), and the minimum required performances for the compressive strength, bending strength, residual compressive strength after 100 cycles of freezing and thawing, skid resistance, and permeability coefficient of permeable concrete are presented. In this study, the compressive strength, bending strength, residual compressive strength after 100 cycles of freezing and thawing, skid resistance, and permeability coefficient were tested to assess the performance of the permeable concrete containing a mixture of active loess and zeolite.

The effect of the method of compaction and compaction intensity of permeable concrete on its strength properties and durability are significant. Therefore, this study compacted permeable concrete by performing the free fall of a rammer that is 50 mm in diameter and 2.5 kg in mass at 300 mm height, as shown in Figure 3. The number of compaction layers and compaction frequency for each layer followed the presented method in the SPS-F-KSPIC-001-2006 standard. Table 7 shows the size of specimen for each test, number of compaction layers, and the compaction frequency for each layer.

The compressive strength of permeable concrete was assessed on days 7, 14, and 28 using a single-axis compression test according to the KS F 2405 standard. The bending strength, skid resistance, and permeability coefficient of permeable concrete were assessed on day 28 according to the KS F 2408, KS F 2375, and KS F 4001 standards, respectively. Lastly, the freeze–thaw resistance was assessed on day 28 using a single-axis compression test according to the KS F 2405 standard with the specimen that underwent 100 cycles of freezing and thawing according to the KS F 2456 standard.

#### 2.3.2. NO_x_ Reduction Performance Evaluation Test

To assess the NO_x_ removal performance of the photocatalyst, the schematic of nitrogen oxide reduction evaluation system that complies with ISO 22197-1(2016) is shown in Figure 4 [28]. The NO_x_ reduction evaluation system consists of a test gas supplier, a photoreactor (test chamber), and a test gas analyzer. The test gas supplier consists of a flow controller, a humidifier, a gas mix tank, etc. A mixture of dry air, moist air that has passed through the humidifier, and polluted gas supplies 1ppm polluted gas with 50% relative humidity. The test chamber for testing NO_x_ reduction was fabricated in two sizes: an ISO standard photoreactor (sample size 100 mm × 50 mm × 5 mm) for small specimens and a tank photoreactor (tank size 240 mm × 140 mm × 200 mm) for large specimens such as concrete and asphalt construction materials. The upper part of the chamber was composed of tempered glass that ultraviolet (UV) could penetrate. As the size of the specimen for testing NO_x_ reduction, a small specimen of 50 mm × 10 mm × 5 mm in size is used for the ISO standard photoreactor, but the ISO standard photoreactor specimen guideline was corrected to the size of the aggregate used in permeable concrete fabrication (10 mm or above), and permeable concrete 50 mm × 10 mm × 25 mm in size was fabricated for use. In the case of the tank photoreactor, a permeable concrete compressive strength test piece (100 mm diameter × 50 mm height) was used. Images of each chamber and specimens are shown in Figure 5 (ISO standard photoreactor) and Figure 6 (tank-type photoreactor). For specimen preprocessing, all specimens were soaked in distilled water for 2 h or above and dried at 40 °C before testing. The preprocessed specimens were coated with a photocatalyst using an automatic spray device. As experimental pollutant gas, NO gas and air were mixed, and 1 ppm NO gas with 50% relative humidity at 25 °C was used. The specimen coated with photocatalyst was located within the chamber and pollutant gas was injected at a flow rate of 3 L min^−1^ and stabilized for 1 h. Then, the pollutant gas concentration was documented for 5 h while irradiating with a UV lamp at a wavelength of 325 nm. Lastly, the UV lamp was turned off and the pollutant gas concentration was observed for 1 h. The NO_x_ reduction efficiency of the photocatalyst could be determined using the NO_x_ concentration (NO_xequil_) at equilibrium during the photocatalyst reaction and the initial NO_x_ concentration (NO_xinitial_) (Equation (2)).
(2)$${\mathrm{NO}}_{x}\;\mathrm{reduction}\;\mathrm{Efficiency}\left(\%\right)=\frac{\left({\mathrm{NO}}_{\mathrm{xinitial}}-{\mathrm{NO}}_{\mathrm{xequil}}\right)}{{\mathrm{NO}}_{\mathrm{xequil}}}\times 100$$

#### 2.3.3. Nitrate Assessment

As an indirect measurement method for validating the photodecomposition reaction of nitrogen oxide, the nitrate concentration on the surface of photocatalyst-coated permeable concrete was analyzed [29]. Distilled water was used to dissolve nitrate on the surface of the permeable concrete. To extract nitrate, specimens that completed the NO_x_ reduction experiment (T0, T7.5, and T10) in both reactors were used. Figure 7 shows the water quality analyzer (a) for measuring nitrate concentration and the process of dissolving nitrate in specimens of ISO standard photoreactor (b) and tank-type photoreactor (c). Given the properties of permeable concrete, the distilled water poured onto the surface would be lost downwards; thus, 40 mL of distilled water was poured into a rectangular container (approximately 63 mL for tank photoreactor specimens) and the coated surface of permeable concrete was soaked into the distilled water to dissolve nitrate for 5 min. The collected solution was filtered through a 0.22 μm filer, and nitrate concentration was measured through a water quality analyzer (SPECTROPHOTOMETER, HS-3700, HUMAS, Daejeon, Korea).

## 3. Results and Discussion

### 3.1. Mechanical Properties of Permeable Concretes

Figure 8 shows the compressive strength test results of permeable concrete according to the number of curing days. The CTRL sample, without the active loess and zeolite mixture, achieved an average compressive strength of 21.64 MPa after 7 days of curing. It exhibited 96.3% of the compressive strength of 22.47 MPa after 28-day curing. In contrast, after curing for 7 days, the compressive strength of A1Z2, A1Z1, and A2Z1, containing a mixture of active loess and zeolite, was 16.74, 17.95, and 15.75 MPa, respectively; on average, they exhibited 84.3% of the 28-day compressive strength after curing. The compressive strength of A1Z2, A1Z1, and A2Z1 at day 28 was 18.95, 20.72, and 20.23 MPa, respectively, which were 84.33%, 92.21%, and 90.03% compared to the CTRL, respectively. In this recipe, the compressive strength of concrete in which some of the cement was replaced with pozzolanic materials (activated loess, natural zeolite) showed a decrease, and when the content of zeolite was higher than that of activated loess, the compressive strength of concrete was greatly reduced. It is judged that when cement is replaced with pozzolanic materials such as zeolite and activated loess, the absolute amount of cement clinker decreases and the pozzolanic reaction of pozzolanic materials such as activated loess and zeolite does not sufficiently proceed. In addition, it is judged that the content of constituent minerals that affect the development of strength, such as SiO_2_, is also reduced [30,31,32,33]. However, the average compressive strength of A1Z2, A1Z1, and A2Z1 after a curing period of 28 days satisfied the quality standard for permeable concrete for road pavement presented by SPS-F-KSPIC-001-2006.

Figure 9 demonstrates the results of the flexural strength test for each specimen after a curing period of 28 days. After curing for 28 days, the flexural strength of the CTRL was 3.88 MPa, while those of A1Z2, A1Z1, and A2Z1, which contain a mixture of zeolite and active loess, were 3.01, 3.63, and 3.02 MPa, respectively. They were, on average, 82.9% of that of the CTRL. Similar to the compressive strength experiment results, the A1Z1 specimen showed high strength compared to A1Z2 and A2Z1. All specimens satisfied the flexural strength standard of permeable concrete for road pavements presented by SPS-F-KSPIC-001-2006. The evaluation results of compressive strength and flexural strength of permeable concrete specimens are summarized in Table 8.

### 3.2. Durability of Permeable Concretes

Table 9 summarizes the permeability coefficient and skid resistance assessment results. The average skid resistance of specimens CTRL, A1Z2, A1Z1, and A2Z1 is 48.3, 50.3, 48, and 49.7 BPN, respectively. All specimens satisfied the standard of 30 BPN for sidewalk concrete and 40 BPN for bicycle road and parking lot concrete presented by SPS-F-KSPIC-001-2006. Therefore, the skid resistance was found to not be significantly affected by the binder type. In contrast, the permeability coefficient was the greatest in A1Z2 at 6.93 × 10^−3^ cm/s with the lowest compressive and flexural strengths. However, this was determined not to be a significant difference between other specimens. The skid resistance of all specimens satisfied the standard of 1 × 10^−3^ cm/s of permeable concrete presented by SPS-F-KSPIC-001-2006.

Figure 10 displays the specimens subjected to the freeze–thaw action of the manufactured permeable concretes. CTRL and A1Z1 did not show significant differences before and after 100 cycles of freezing and thawing, as shown in Figure 10a,c. In contrast, as shown in Figure 10b,d, the surface aggregate was severely eliminated, and cracks were observed. Therefore, residual compressive strength testing was performed for CTRL and A1Z1, excluding A1Z2 and A2Z1, after completing the freezing and thawing testing, and the results are summarized in Table 10. The test results showed that, on average, the residual compressive strengths of CTRL and A1Z1 were 19.6 and 16.8 MPa, respectively, which are 87.1% and 82.4% of the compressive strength on day 28, respectively. Thus, they were found to satisfy 80% or above the SPS-F-KSPIC-001-2006 standard. Therefore, in subsequent studies, A1Z1 specimens that satisfied all permeable concreate evaluation criteria were used.

### 3.3. NO_x_ Reduction Performance Evaluation According to TiO_2_ Content

To assess the nitrogen oxide removal performance of the photocatalyst according to TiO_2_ content, the photocatalyst was coated with varying TiO_2_ contents onto the A1Z1 permeable concrete supporter, and a nitrogen oxide removal experiment was performed in the ISO standard photoreactor chamber according to ISO 22197-1:2016.

Figure 11 shows the results of assessing NO_x_ reduction performance according to photocatalyst content, and the average NO_x_ reduction efficiency values of each specimen are summarized in Table 11. In A1Z1 (T0) without photocatalyst coating, it was found that photodecomposition reaction was nonexistent with NO_x_ reduction efficiency of 1.6%. In all specimens with photocatalyst coating, the NO gas concentration rapidly decreased owing to the activation of the photocatalyst reaction after lighting the UV lamp, the photocatalyst reaction was deactivated after turning off the UV lamp, and recovery to the initial concentration was shown. Additionally, the tendency of an increase in nitrogen oxide removal efficiency was shown according to the increase in photocatalyst content, and the NO_x_ reduction effect was the greatest at 77.5% with 7.5% TiO_2_ content. Meanwhile, with 10% TiO_2_ content, the NO_x_ reduction efficiency was 67.7%, which was lower than that with 7.5% TiO_2_ content despite the higher TiO_2_ content. In this result, the reason for the maximum NO_x_ reduction effect of permeable concrete using T7.5 photocatalyst is that TiO_2_ particles of 70 nm show the most appropriate dispersion and adsorption at a content of 7.5% in macro- and micropores on the surfaces of active loess and zeolite as porous materials and coarse aggregate.

For the NO_x_ reduction assessment of construction materials with large volumes, such as concrete and asphalt, a tank photoreactor chamber that modified the ISO 22197-1:2016 standard experiment regulation was fabricated, and the NO_x_ reduction experiment was performed with the compressive strength test piece. For the photocatalyst used in surface coating, T7.5 and T10, which demonstrated high efficiency in previous experiments, were used, and the results are shown in Figure 12 and Table 12. The NO_x_ reduction efficiency value in the sample using the compressive strength test piece in the tank photoreactor was lower than that in the ISO standard photoreactor despite the broad coating area. The observed low NO_x_ reduction efficiency can be attributed to the larger volume of the tank photoreactor compared to the specimen in the ISO standard photoreactors. As the volume inside the chamber increased, the NO_x_ reduction efficiency decreased due to the increase in the amount of pollutants to be reduced compared to the photocatalyst coating unit area. In addition, as shown in Figure 12, higher efficiency was shown with 7.5% TiO_2_ content than with 10% TiO_2_ content.

### 3.4. Nitrate Analysis after Photolysis of Nitrogen Oxides

As an indirect method of identifying the TiO_2_ photocatalyst decomposition reaction, the concentration of nitrate generated directly after the NO_x_ reduction experiment was analyzed. The nitrate content accumulated on the surface of T0, T7.5, and T10 specimens that completed the NO_x_ reduction experiment from the ISO standard photoreactor and tank photoreactor was dissolved in distilled water for extraction. Nitrate concentrations in the specimens in both photoreactor types measured through the water quality analyzer are summarized in Table 13. Compared to T0, which did not undergo photocatalyst treatment, nitrate concentration measurement following the NO_x_ reduction experiment revealed that a photodecomposition reaction occurred in the specimen treated with TiO_2_ photocatalyst. In addition, higher nitrate concentration was shown in specimen T7.5 than in T10; this was attributed to the greater amount of nitrate generated in specimen T7.5 with higher NO_x_ reduction efficiency than in T10, as shown in the NO_x_ reduction experiment results. The specimens in the tank photoreactor exhibited a similar trend to the results observed in the ISO standard photoreactor specimens.

## 4. Conclusions

The application of TiO_2_ photocatalyst to construction materials for enhancing air quality has garnered significant interest among researchers in both industry and academia. In line with the objective of creating functional construction materials to reduce NO_x_ emissions from roadside sources, this study focused on developing permeable concrete containing porous materials like active loess and zeolite for sidewalk pavements. The NO_x_ reduction performance was then evaluated with the application of TiO_2_. Based on our findings, the following conclusions can be drawn:1.A decrease in the physical performance of permeable concrete that includes active loess and zeolite was observed in comparison to the CTRL, but they all satisfied the quality standard of permeable concrete for sidewalk and road pavement.2.The skid resistance and permeability coefficient showed results that satisfy standards for sidewalk concrete in all permeable concrete specimens, but the quality standard was satisfied at 80% or above of residual compressive strength in only CTRL and A1Z1 specimens in the freezing and thawing experiment.3.Considering the physical and durability performances of this study, A1Z1 mixture was determined as the optimal mixing ratio for permeable concrete for applying TiO_2_ photocatalyst.4.The NO_x_ reduction efficiency tended to increase according to the increase in TiO_2_ content. With 7.5% TiO_2_ content, a maximum NO_x_ reduction efficiency of 77.5% was observed.5.The tank photoreactor of the specimen using the compressive strength test piece also showed similar results to the NOx reduction tendency in the ISO standard photoreactor.6.By assessing the nitrate concentration generated after NO_x_ reduction assessment, it was found that a photocatalyst reaction occurred during UV irradiation on the surface of the photocatalyst-coated permeable concrete. Additionally, higher nitrate concentration and higher NO_x_ reduction efficiency were observed with 7.5% TiO_2_ content than with 10% TiO_2_ content.

## Figures and Tables

**Figure 1 materials-16-05512-f001:**
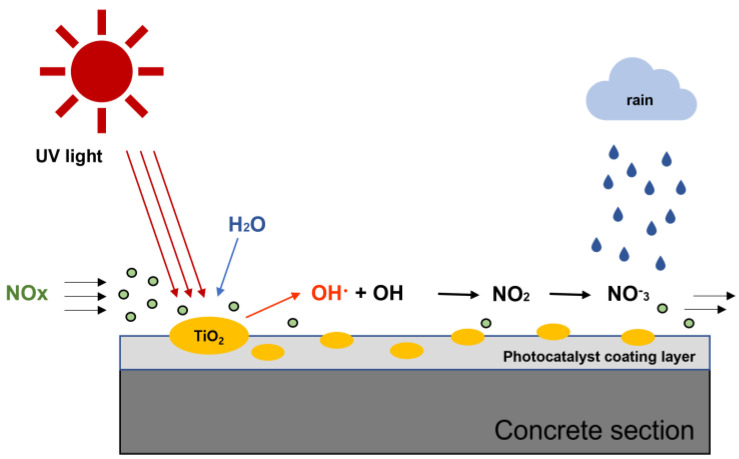
Schematic representation of NO_x_ decomposition via TiO_2_ photocatalytic reaction.

**Figure 2 materials-16-05512-f002:**
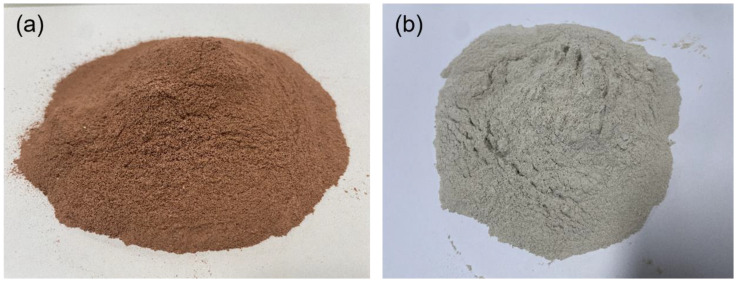
Images of (**a**) active loess and (**b**) zeolite.

**Figure 3 materials-16-05512-f003:**
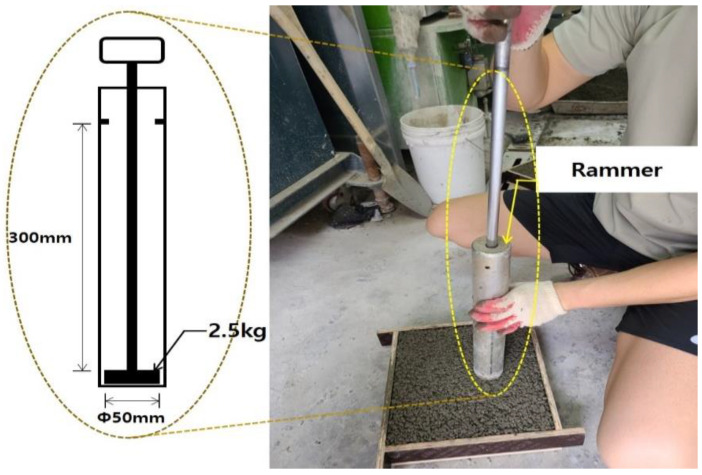
An overview of specimen casting.

**Figure 4 materials-16-05512-f004:**
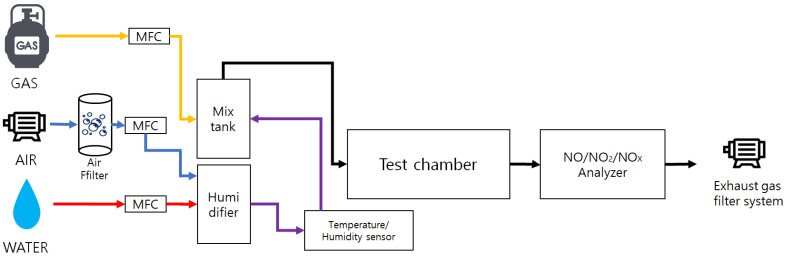
Schematic diagram of the NO_x_ reduction performance assessment system.

**Figure 5 materials-16-05512-f005:**
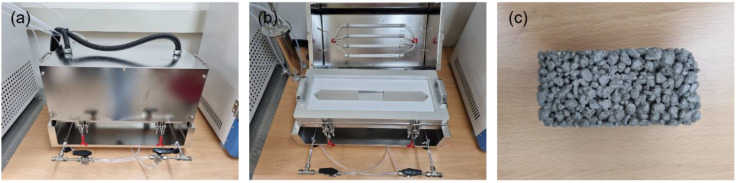
Images of the ISO standard photoreactor: (**a**) exterior, (**b**) interior, and (**c**) specimen.

**Figure 6 materials-16-05512-f006:**
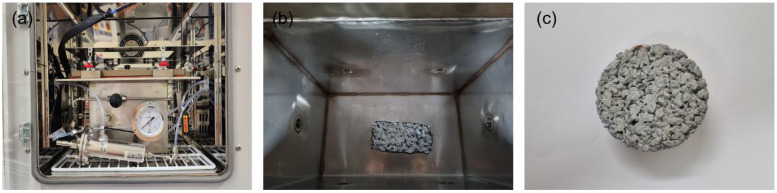
Images of the tank-type photoreactor: (**a**) exterior, (**b**) interior, and (**c**) specimen.

**Figure 7 materials-16-05512-f007:**
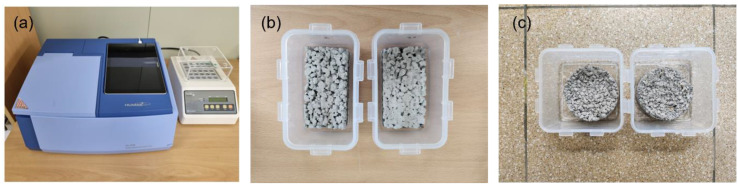
Images of nitrate concentration measurement equipment and nitrate collection of specimens: (**a**) water quality analyzer; (**b**) specimens of ISO standard photoreactor; (**c**) specimens of tank-type photoreactor.

**Figure 8 materials-16-05512-f008:**
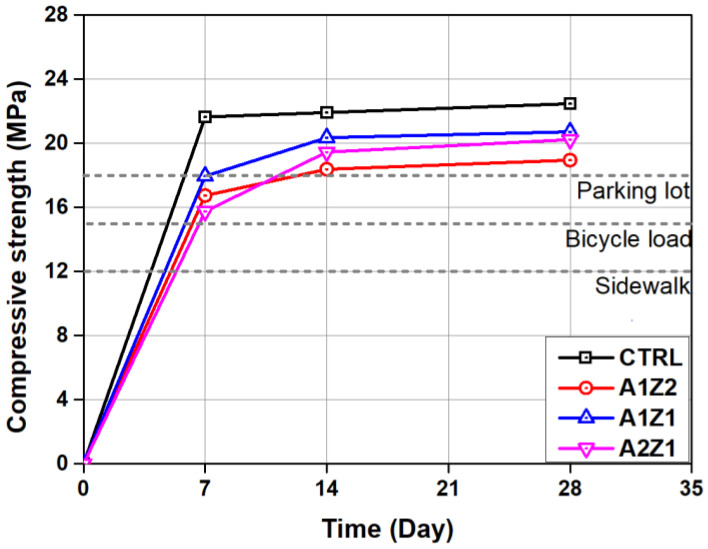
Results of compressive strength of permeable concrete specimens.

**Figure 9 materials-16-05512-f009:**
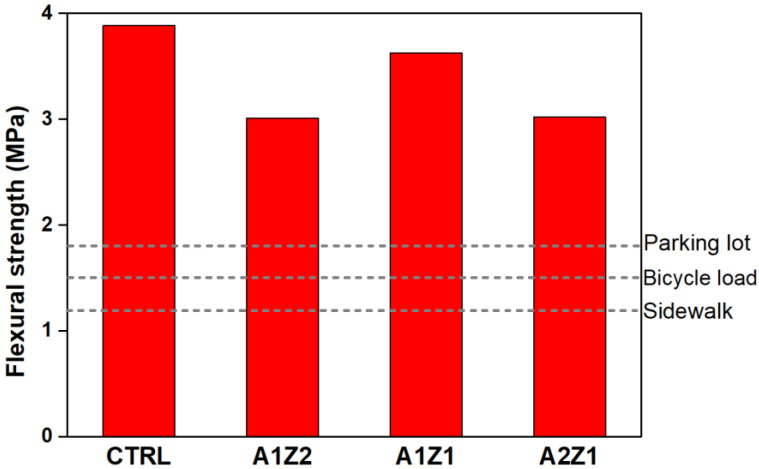
Results of flexural strength of permeable concrete specimens.

**Figure 10 materials-16-05512-f010:**
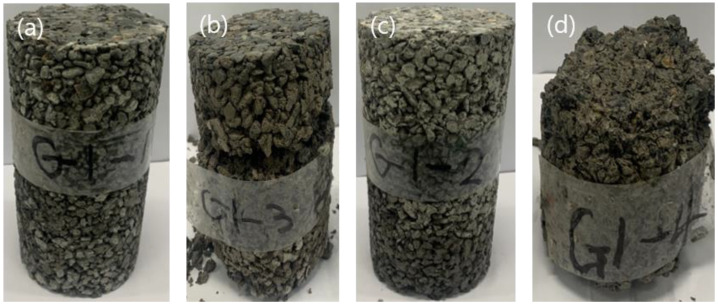
Image of the specimens following the freezing and thawing test. (**a**) CTRL, (**b**) A1Z2, (**c**) A1Z1, and (**d**) A2Z1.

**Figure 11 materials-16-05512-f011:**
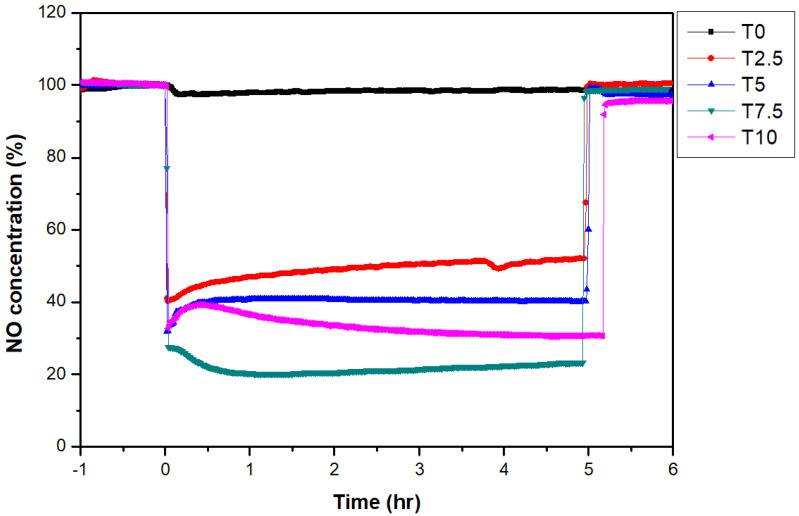
Changes in NO concentration according to TiO_2_ content.

**Figure 12 materials-16-05512-f012:**
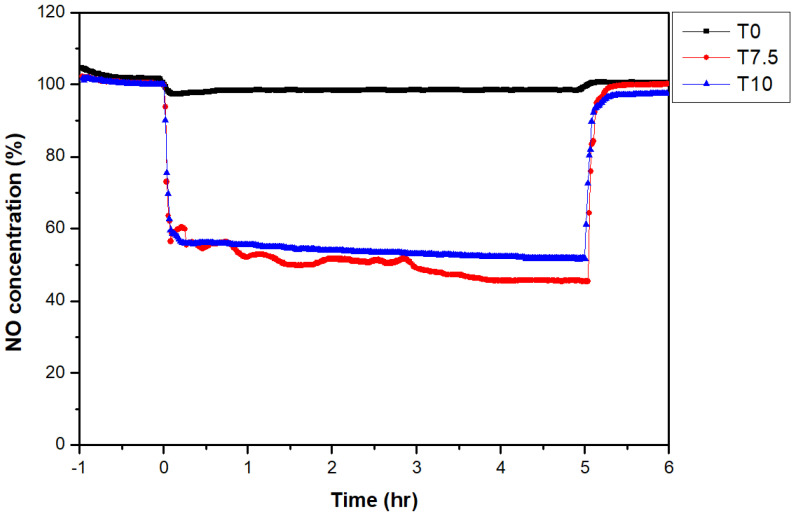
Changes in NO concentration according to the TiO_2_ content in the tank photoreactor.

**Table 1 materials-16-05512-t001:** Chemical composition of cement.

CaO	SiO_2_	Al_2_O_3_	Fe_2_O_3_	MgO	SO_3_	Etc.
62.79%	21.74%	5.00%	3.17%	2.97%	1.67%	1.37%

**Table 2 materials-16-05512-t002:** Chemical composition of active loess and zeolite.

	SiO_2_	Al_2_O_3_	Fe_2_O_3_	CaO	MgO	K_2_O	Etc.
Active loess	43.0%	35.9%	10.8%	7.2%	1.6%	0.8%	1.7%
Zeolite	68.9%	16.4%	5.3%	2.6%	1.0%	3.7%	2.1%

**Table 3 materials-16-05512-t003:** Physicochemical properties of the TiO_2_ photocatalyst.

Properties	Unit	Value
TiO_2_ type	-	Anatase
TiO_2_ content	%	39–41
Viscosity	MPa·s	≤30
pH value	-	5.0–7.0
Density at 20 °C	g/m^3^	1.41
Particle size	nm	70

**Table 4 materials-16-05512-t004:** Mix proportions of permeable concrete.

Specimen	W/B (%)	Water (kg)	Binder (kg)			Gravel (kg)
			Cement	Active Loess	Zeolite	
CTRL	36.4	120	330	-	-	1600
A1Z2			280	16.5	33	
A1Z1				24.75	24.75	
A2Z1				33	16.5	

**Table 5 materials-16-05512-t005:** Mix proportions of the diluted photocatalyst.

Photocatalyst Code	T0	T2.5	T5	T7.5	T10
AERODISP^®^W740X (g)	-	6.2	12.5	18.8	25.0
DI water (g)	-	93.8	87.3	81.2	75.0
TiO_2_ content (%)	0	2.5	5	7.5	10

**Table 6 materials-16-05512-t006:** Evaluation criteria and quality requirement of permeable concrete for pavement.

	Sidewalk	Bicycle Load	Parking Lot	Standard
Compressive strength (MPa)	12 or more	15 or more	18 or more	KS F 2405
Flexural strength (MPa)	1.2 or more	1.5 or more	1.8 or more	KS F 2408
Compressive strength after 100 cycles of freezing and thawing (%)	At least 80% of strength at 28 days			KS F 2456 KS F 2405
Skid resistance (BPN)	30 or more	40 or more	40 or more	KS F 2375
Permeability coefficient (cm/s)	1.0 × 10^−3^			KS F 4001

**Table 7 materials-16-05512-t007:** Specimen size and compaction method details.

	Specimen Size (mm)	Compaction Layer	Compaction Frequency per Layer
Compressive strength	ϕ100 × 200	3	25
Flexural strength	150 × 150 × 530	2	80
Compressive strength after 100 freeze–thaw cycles	ϕ100 × 200	3	25
Skid resistance	150 × 90 × 50	1	14
Permeability coefficient	300 × 300 × 60	1	90

**Table 8 materials-16-05512-t008:** Compressive strength and flexural strength of permeable concrete specimens.

Sample	Compressive Strength (MPa)			Flexural Strength (MPa)
	7 Days	14 Days	28 Days	28 Days
CTRL	21.64	22.04	33.47	3.88
A1Z2	16.74	18.12	28.95	3.01
A1Z1	17.95	20.35	20.72	3.63
A2Z1	15.75	19.56	20.23	3.02

**Table 9 materials-16-05512-t009:** Skid resistance and permeability coefficient of permeable concrete specimens.

Item	NO.	CTRL	A1Z2	A1Z1	A2Z1
Skid resistance (BPN)	T1	46	51	46	51
	T2	50	52	48	48
	T3	49	48	50	50
	Average	48.3	50.3	48.0	49.7
Permeability coefficient (×10^−3^ cm/s)	T1	7.0	7	6.1	6.3
	T2	6.9	7.3	6.3	6.2
	T3	6.4	7.5	5.9	6.6
	Average	6.77	6.93	6.1	6.37

**Table 10 materials-16-05512-t010:** Freezing and thawing resistance of permeable concrete specimens.

Sample Code	Compressive Strength (MPa)		Freezing and Thawing Resistance (%)
	Before	After	
CTRL	22.47	19.6	87.1
A1Z2	18.95	-	-
A1Z1	20.72	16.8	82.4
A2Z1	20.23	-	-

**Table 11 materials-16-05512-t011:** NO_x_ reduction efficiencies according to TiO_2_ content.

Sample Code	Efficiency of NO_x_ Reduction (%)
T0	1.6
T2.5	49.8
T5	59.6
T7.5	77.5
T10	67.7

**Table 12 materials-16-05512-t012:** NO_x_ reduction efficiencies according to the TiO_2_ content in the tank photoreactor.

Sample Code	Efficiency of NO_x_ Reduction (%)
T0	1.4
T7.5	51.2
T10	46.9

**Table 13 materials-16-05512-t013:** Nitrate concentrations of NO_x_ reduction specimens.

Sample Code	Nitrate Concentration (mg/L)			
	ISO Standard Photoreactor		Tank Photoreactor	
	1st	2nd	1st	2nd
T0	0.01	0.01	0	0.1
T7.5	2.36	2.28	2.16	2.53
T10	1.72	1.69	1.89	2.26

## Data Availability

The data are available in a publicly accessible repository.

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
