# Peer review of "Evaluation of Nitrogen Oxide Reduction Performance in Permeable Concrete Surfaces Treated with a TiO2 Photocatalyst"

_materials, 2023, doi:10.3390/ma16165512_

Round 1

Reviewer 1 Report

The manuscript entitled (Evaluation of Nitrogen Oxide Reduction Performance in Permeable Concrete Surfaces Treated with a TiO2 Photocatalyst) by Hyeokjung et al is good article, however, the present research article aims to evaluate the nitrogen oxide reduction performance in permeable concrete surfaces treated with a TiO2 photocatalyst. This study represents a significant effort by the authors, reflecting their deep understanding of the subject matter. However, several comments have been identified that could further enhance the quality and comprehensiveness of this work.:

1.    The author has mentioned in the abstract that all samples met the quality standards according to ISO standards. However, it would be beneficial to specify the ISO standard number for more clearness.

2.    It is recommended to include the removal efficiency percentage of NOx in the abstract to provide a comprehensive overview of the study findings.

3.    Authors should be more creative in their choice of keywords. The authors used practically the same words as in the title.

4.    The experimental results regarding compressive strength, bending strength, skid resistance, and permeability coefficient of permeable concrete, along with their respective standard limits, should be presented in Table 8.

5.    Lines 224-225 present a textual description of Figure 4 that does not accurately correspond to the actual representation and caption of Figure 4, this inconsistency should be addressed and rectified.

6.    The caption of Figure 7 would benefit from more detailed descriptions regarding the subfigures a, b, and c to provide a clearer understanding of the content presented.

7.    In lines 281 and 286, the repetition of the sentence "The reason for this was attributed to the active loess and zeolite being pozzolan material as described above" should be revised and improved for better clarity and cohesion.

8.    The absence of a textual description for Table 10 is noted. It is recommended to provide a comprehensive explanation or summary of the contents and implications of Table 10 within the main text.

9.    There is an error in line 371, where "T.75" should be corrected to "T7.5," and the word "swere" should be rectified for accuracy.

10. In lines 375, 377, 393, and 404, the term "Nox" should be corrected to "NOx" to maintain consistency and adhere to standard scientific notation.

11. In line 380, it is advised to include a comment on the unexpected results of the superior NOx removal efficiency of T7.5 compared to T10, rather than simply mentioning it. Providing a brief analysis or speculation about these findings would enhance the discussion.

12.  In the Introduction part you can cite the following papers regarding the TiO2 as a photocatalyst:

Silver decorated TiO2/g-C3N4 bifunctional nanocomposites for photocatalytic elimination of water pollutants under UV and artificial solar light

Reviewer 2 Report

Dear authors,

You have done a good work on "Evaluation of Nitrogen Oxide Reduction Performance in Permeable Concrete Sur-faces Treated with a TiO2 Photocatalyst". The following corrections may be done.

1) The number of articles you refereed are very less.

2) The introduction and literature review part can be improved and you can make it better.

3) The technical language can be modified with an expert

4) The conclusion part can be made clearer. 

5) The figures and tables are good.

6) This seems like a track changes version. When you resubmit, you need to upload proper files.

Reviewer 3 Report

 In this manuscript, the authors investigated the NOx reduction performance of road pavement concrete coated with TiO2 photocatalysts. This manuscript can be published in this journal after some major revisions.

1. The Abstract needs great improvement. The abstract should be brevity, clarity, generality, self-contained. The innovation points are not clearly stated. Moreover, the significance of this study should be mentioned. 

2. Please further emphasize the necessity and innovation of this material research in Introduction.

3. The manuscript has lots of format errors or informal expressions. Some errors selected from the paper as follows:

(1) ‘ Mpa’ should be ‘MPa’;

(2) ‘ Nox reduction’ should be ‘NOx reduction’ in Section 3;

(3) ‘ T.7.5’ and ‘T.75’ should be ‘T7.5’ in Section 3;

.........

Please revise your manuscript throughoutly.

4. The NOx decomposition process should be analyzed in detail in Section 3, not in Introduction.

5. The mechanism leading to the improvement of NOx reduction efficiency is not clear.

Minor editing of English language required.

Round 2

Reviewer 1 Report

I would like to thank all authors for their effort to improve the quailty of the manuscript.

Reviewer 2 Report

Paper can be accepted

Reviewer 3 Report

The authors have replied my comments, the quality of this manuscript is improved, so I suggest its publication in Materials.